# The Application of Olive-Derived Polyphenols on Exercise-Induced Inflammation: A Scoping Review

**DOI:** 10.3390/nu17020223

**Published:** 2025-01-09

**Authors:** Joseph B. Lillis, Ashley G. B. Willmott, Havovi Chichger, Justin D. Roberts

**Affiliations:** 1Cambridge Centre for Sport and Exercise Sciences (CCSES), Faculty of Science and Engineering, Anglia Ruskin University, Cambridge CB1 1PT, UK; ash.willmott@aru.ac.uk (A.G.B.W.); justin.roberts@aru.ac.uk (J.D.R.); 2School of Life Sciences, Faculty of Science and Engineering, Anglia Ruskin University, Cambridge CB1 1PT, UK; havovi.chichger@aru.ac.uk

**Keywords:** inflammation, exercise, polyphenols, olives, hydroxytyrosol

## Abstract

Background/Objectives: There is current scientific interest pertaining to the therapeutic effects of olive-derived polyphenols (ODPs), in particular their associated anti-inflammatory properties, following the wealth of research surrounding the physiological impact of the Mediterranean Diet (MD). Despite this association, the majority of the current literature investigates ODPs in conjunction with metabolic diseases. There is limited research focusing on ODPs and acute inflammation following exercise, regardless of the knowledge surrounding the elevated inflammatory response during this time. Therefore, the aim of this scoping review is to understand the impact ODPs may have on exercise-induced inflammation. Methods: This scoping review was undertaken in accordance with the Preferred Reporting Items for Systematic Reviews and Meta-Analyses extension for Scoping Reviews (PRISMA-ScRs). The literature searches were conducted in PubMed and EBSCOhost and considered for review if records reported original data, examined olives, olive-derived nutrients, food sources, or ODPs in conjunction with exercise-induced inflammation (including known causes, associations, and proxy measures). Results: Seven studies investigated ODPs and exercise-induced inflammation, providing commentary on reduced oxidative stress, inflammatory biomarkers, and immune biomarkers, enhanced antioxidant defenses and modulations in mitochondrial function, albeit in low numbers. An average of 100.9 mg∙d^−1^ ODPs were supplemented for an average of 40 days, with hydroxytyrosol (HT) being the primary ODP investigated. Six studies employed individual aerobic exercise as their stimulus, whilst one study investigated the impact of an acute dose of ODP. Conclusions: There is a limited consensus on the direction of isolated HT in human models, whereas animal models suggest a reduced inflammatory response following ≥2 weeks HT supplementation in conjunction with chronic exercise. Future research should initially investigate the inflammatory response of ODP, with particular focus on HT, and aim to identify an optimum dose and time course for supplementation surrounding exercise to support acute recovery and exercise adaptations.

## 1. Introduction

Following the wealth of investigation into the physiological impact of the Mediterranean Diet (MD) [1], there is current experimental, clinical, and epidemiological research demonstrating the potent therapeutic effects of olive-derived polyphenols (ODPs) [2]. To date, the majority of this evidence focuses on the impact of olive-derived bioactive compounds on metabolic diseases and neurodegenerative disorders, with well-documented examples of neuroprotection [3], cardio-metabolic protection [4], anti-tumor and anti-inflammatory properties [5]. The discernible health benefits associated with the MD, notorious for high polyphenol intakes, in some instances exceeding 3000 mg∙d^−1^ [6,7], have been partially attributed to the consumption of olives and olive-derived products [8]. Primarily, olives and olive oil comprise saturated (in the form of glycerol esters), poly-unsaturated, and mono-unsaturated fatty acids in addition to a variety of vitamins and minerals (Vitamin K, tocopherols, beta-carotene, calcium, iron, and potassium) in different quantities and ratios depending upon the strain and harvest of olive fruit [9]. In addition to tocopherols (Vitamin E), various other antioxidants and polyphenols are present within olives, olive leaves, and olive oil [10]. The most potent concentration is provided by secoiridoids such as oleacein, oleocanthal, and oleuropein aglycone, however lignans, flavones and phenolic acids, and alcohols are also present and may interact to elicit physiological effects, although concentrations vary greatly between crop, strain, and part of the fruit [10,11]. It has been stipulated that ODPs including phenolic acids and related derivatives such as hydroxytyrosol (HT (Chemical formula: C_8_H_10_O_3_, molar mass: 154.17 g·mol^−1^)) and oleuropein (Ole (Chemical formula: C_25_H_32_O_13_ molar mass: 540.52 g·mol^−1^)) [10] found in olive fruit, leaves, unrefined virgin oil, and mill wastewater [12] have the ability to sustain, support, and stabilize the endogenous antioxidant system at a cellular level through inhibiting mitochondrial dysfunction, attenuating lipid peroxidation, and protecting against oxidative stress via radical scavenging [13,14,15,16]. In most cases, HT is extracted from olive leaves via the following four-step process: increasing the olive leaf area (pre-treatment), solid-liquid extraction, hydrolysis reaction, and pH adjustment [17]; however, other methods have been described elsewhere [18]. Despite being recognized as some of the primary ODPs, there is limited research pertaining to the effects of HT or Ole in an exercise domain, and a paucity of evidence surrounding the absorption of olive-derived phenolics and their potential in vivo activity. However, it is understood that ODPs undergo rapid absorption, primarily via the colonic microflora [19], followed by extensive metabolism (after absorption through the gut barrier) either within tissue or via the colonic microflora (for the small fraction not initially absorbed or re-excreted in bile [19]). In the case of HT, metabolism involves an initial conversion of HT into oxidized or methylated derivatives, necessary to accommodate uptake through the gastrointestinal lumen, and then secondly, a transformation process (e.g., homovanillic alcohol catalyzed into homovanilic acid via catechol-*O*-methyltransferase) that leads to the formation of metabolites, such as sulphate and glucuronide conjugates, in some cases, ~5 to ~30 min following ingestion [20,21,22]. Therefore, it could be suggested that similar to the natural environment, whereby secondary metabolites are produced by plants to capitalize and compete in their surroundings, the presence of circulating secondary metabolites may support the operational capacity of ODP in the human body. This could play an important role in exerting the proposed positive physiological impacts of ODP (e.g., antioxidant, antimicrobial, and anti-inflammatory [23]).

Despite the limited literature in human models, various animal studies have recognized that certain ODPs, such as HT or Ole, are particularly effective at combatting inflammation [24]. This likely occurs via inhibiting the nuclear factor-kappa B (NF-κB) pathway, responsible for inducing the expression of pro-inflammatory mediators [25], and importantly, activating the nuclear factor erythroid 2-related factor 2 (Nrf2) pathway [24], the key regulator of redox homeostasis with the human body [26]. From an exercise perspective, the disturbed redox balance present following exercise-induced oxidative stress is a direct result of reactive oxygen species (ROS) levels exceeding endogenous antioxidant capacity. The result of this imbalance can cause an inflammatory response, such as increases in interleukins (IL) IL- 1β, IL-6, and IL-10, tumor necrosis factor alpha (TNF-α), or peroxisome proliferator-activated receptor gamma (PPAR-γ) [27]), which may impair the functionality of the affected tissue (e.g., loss of muscle strength, power, or range of motion surrounding the affected joint) [27]. The endogenous antioxidant system (including enzymes super oxide dismutase [SOD], catalase [CAT], and reduced glutathione [GSH]) is employed to combat oxidative stress and support the removal of free radicals; however, this process is complex and multifaceted. Importantly, this system can become inefficient if circumstances that evoke high levels of oxidative stress, inflammation, and therefore ROS, such as intense exercise in trained individuals (short bouts (≤20 min) ≥ 85% $\overset{\cdot }{V}$O_2max_ or prolonged exercise (≥20 min) at ~60–84% $\overset{\cdot }{V}$O_2max_) or exercise in an untrained population [28], outweigh levels of endogenous antioxidant enzymes. SOD is the first line of defense against ROS following the increased oxygen consumption and subsequent increase in oxidative stress during exercise and physical activity [29]. SOD is fundamental in the detoxification of superoxide radicals, which can be intrinsically reactive, particularly around other reactive radicals, and therefore promote oxygen toxicity and have a negative impact on the structure of the cell [30]. Through interacting with the activity of various proteins, it is understood that ROS can significantly impact cellular function [31]. Manganese superoxide dismutase (MnSOD) is an inducible form of superoxide dismutase that is primarily located in the mitochondria [32]. Due to this location, MnSOD is considered a fundamental enzyme involved in the detoxification of ROS [31] and through converting superoxide (O_2_^•−^) into molecular oxygen, supports the maintenance and function of the mitochondria [33].

From an exercise perspective, whereby an increase in oxidative stress is synonymous, regardless of the experience or training status of the individual, this mechanism could be important in reducing the time required to recover the muscle during both longer duration single bouts of exercise ((≥20 min) at ~60–84% $\overset{\cdot }{V}$O_2max_) and shorter duration interval-based training ((≤20 min) ≥85% $\overset{\cdot }{V}$O_2max_) and events, e.g., high-intensity interval training, CrossFit, or HYROX (a multidisciplinary functional exercise race). Interestingly, initial research investigating HT and exercise has demonstrated the positive influence that a HT-rich phytocomplex, derived from olive mill wastewater, may pose on parameters of aerobic exercise and subsequent acute recovery at lower exercise intensities (≤~60% $\overset{\cdot }{V}$O_2max_) [34]. These findings corroborate and expand upon the existing knowledge surrounding plant-derived nutrients and exercise performance, and thus warrant further investigation into ODP, such as HT, and their possible application to exercise. The primary aim of this scoping review is therefore to understand the impact that ODP may have on exercise-induced inflammation, and secondly, to formulate empirical evidence to support and direct future recommendations for supplementation within exercise and exercise recovery.

## 2. Materials and Methods

Due to the volume of available research, a scoping review was selected, as opposed to a systematic approach with meta-analyses, to establish a body of literature pertaining to ODP and exercise-induced inflammation to specifically highlight the research question of interest. The current scoping review was completed in line with the checklist for Preferred Reporting Items for Systematic Reviews and Meta-Analyses extension for Scoping Reviews (PRISMA-ScR) [35].

### 2.1. Eligibility

In order to be included for review, research needed to report original data that were published in peer-reviewed academic journals, examine olive or olive-derived nutrients and food sources, and investigate exercise-induced inflammation (including known causes and associations, e.g., exercise-induced oxidative stress, muscle damage, or mitochondrial perturbations). Research was not considered for review if the following applied: (i) a publication occurred before the year 2000, (ii) research was not published in the English language, (iii) full text was not available, (iv) the research was a thesis dissertation, (v) research was conducted on participants in an elderly population (>65 years), or those with chronic and/or metabolic disease states.

### 2.2. Search Strategy

Literature searches were conducted in PubMed and EBSCOhost. Within EBSOCOhost, Psychological, Education, and Business databases were removed prior to the initial literature search, followed by the removal of exact duplicate results. Search and MeSH terms employed are provided in Table 1.

### 2.3. Study Selection and Data Extraction

Following the initial search, all results were screened by the lead researcher (J.L). Initially, titles and abstracts were read, with any non-relevant literature removed, all the remaining literature was collated by the lead researcher, and together with a secondary member of the research team (J.R), screened against the inclusion and exclusion criteria. Importantly, at this stage, all records that included investigation into a disease state were discarded. This strategy was employed to eliminate any reports that investigated chronic inflammation (≥6 months) present with metabolic pathologies, and therefore single out the inflammatory response to exercise, focusing on acute inflammation and importantly, specific cell types, proteins, mechanisms, and symptoms that may be exhibited during or after exercise. In addition, the reference lists of included records were reviewed by the lead researcher in order to identify any pertinent research omitted from the electronic literature search. This process was repeated where necessary until no further records were detected. Following their inclusion, the title, author, year of publication, sample size, study design, intervention method (including dosing strategy if appropriate), exercise protocol, and inflammatory marker(s), mediator(s), or associated mechanisms (where appropriate) were identified, whilst key findings were recorded for later analysis [35].

## 3. Results

### 3.1. Literature Search

The electronic database search provided 478 records; 417 were sourced from PubMed and following the removal of duplicate results (*n* = 17), a further 44 were sourced from EBSCOhost. A total of 426 records were excluded due to non-relevant titles or abstracts and of the 35 remaining records, a further 29 were removed following full document screening due to non-compliance with the pre-determined inclusion and/or exclusion criteria. Moreover, following a review of the remaining literature’s reference lists (*n* = 6), an additional 2 records were deemed relevant and warranted full-text screening for eligibility. Following this process, 1 was excluded due to the investigation involving a clinical population, and therefore, a total of 7 records were included for review (Figure 1).

### 3.2. Synthesis of Results

From the scoping review, hydroxytyrosol appeared to offer the most potent in vitro antioxidant potential of all olive oil polyphenols [36], and therefore, was the primary ODP investigated, utilized in three experimental studies. Olive oil, an olive-derived phytocomplex, olive leaf extract, and maslinic acid each appeared as individual supplementation strategies in the four remaining records. On average, 100.9 mg∙d^−1^ of the total ODP was supplemented (16.4 mg∙d^−1^ in animal studies and 241.7 mg∙d^−1^ in human), and importantly, all records included within this review employed an experimental control. One study investigated an acute (1 h pre-exercise) dose response of ODP [37], meanwhile six studies supplemented for an average of forty days (7–70 days) [34,38,39,40,41,42].

Three studies utilized human participants, with one study focusing on team sport and the remaining two, in addition to all four animal studies, employing individual aerobic exercise as their stimulus. On average, experimental groups in animal studies included thirty rodents, in comparison to an average of fifteen participants in human studies (height: 1.76 m ± 0.2 m, body mass: 76.4 kg ± 0.5 kg, age: 30 year ± 11 years). The reviewed research recorded the impact of ODP via a decrease in inflammatory and immune biomarkers [37,42], enhanced antioxidant defenses [34,40], reduction in oxidative stress [34,37,39,40], and modulated mitochondrial function [38,41], as seen in Table 2. Interestingly, in addition to an increased maximum running velocity, one study reported a pro-inflammatory effect of HT supplementation following a high chronic dose (300 mg∙kg^−1^∙d^−1^ for 10 weeks). Importantly, in all studies reviewed, exercise (evidenced by blood lactate concentration, duration, or visual analogue scales) and diet (evidenced by energy intake and macronutrient split) were matched between groups (where necessary), providing satisfactory evidence of an appropriate control of study parameters.

## 4. Discussion

The primary aim of this review was to identify whether ODPs can impact exercise-induced inflammation. Additionally, a secondary aim was to formulate the data within this area of research to support and direct future recommendations for supplementation within exercise and exercise recovery. Hydroxytyrosol was the most widely investigated ODP [38,39,40], and in some cases, solely attributed to the positive influence on exercise performance [34]. Interestingly, authors suggest that an increase in mitochondrial biogenesis, detectable via the activated peroxisome proliferator-activated receptor-gamma coactivator (PGC-1α) and a reduced expression of NF-κB, may be responsible for the elicited physiological impact(s) on inflammation following ODP supplementation. The following section of this review therefore considers the mechanistic actions of ODPs in both animal and human models, specific to inflammation.

### 4.1. Animal Studies

Interestingly, all four animal studies included in this review employed chronic supplementation for between five and ten weeks (an average 54 days) in comparison to an average of 11 days in human studies (see Table 2). Although the reported physiological permutations and improvements in exercise performance following chronic supplementation [38,39,40,41] are on the surface encouraging, there are two noteworthy criticisms of the reviewed literature. Firstly, the isolated use of polyphenols was not comprehensively investigated in all experimental groups; therefore, conclusions can only truly be based upon a relative model that includes regular exercise training and/or dietary controls. Subsequent improvements in endurance capacity [40,41] could thus be attributed to the regular and consistent endurance exercise training that elsewhere has been reported to improve mitochondrial biogenesis in the absence of supplementation [43]. Secondly, there is limited consistency within the manner in which change in mitochondrial biogenesis is evaluated. Three studies in the current review isolate mitochondria from the tissue [38,39,40] in addition to measuring secondary indicators of mitochondrial function, such as the mRNA expression levels of the Takeda G protein-coupled receptor 5 (TGR5), PGC-1α, Sirtuin 1 (Sirt1), and Forkhead box O3. Although these markers provide an excellent indication of the trajectory of mitochondrial biogenesis, and by virtue inflammatory status, in the absence of measuring the synthesis rates of mitochondrial proteins, transcriptional factors for mitochondrial genes, and the mRNAs encoding mitochondrial proteins, they do not provide a comprehensive evaluation of mitochondrial biogenesis. Nevertheless, the increase in mitochondrial biogenesis, mitochondrial mass in the muscle, and therefore mitochondrial capacity for aerobic production can be attributed to the activation of PGC-1α, which various polyphenols have the capacity to influence, subsequently indicating a suppression of inflammation [44]. The exhibited increase in mitochondrial biogenesis is supported by the increase in mitochondrial complex I and II following eight weeks of chronic HT supplementation [40]. This has a positive down-stream impact on oxidative phosphorylation and adenosine triphosphate (ATP) production due to the increased efficiency of the mitochondrial electron transport chain [40]. Interestingly, Al Fazazi et al. (2018) [39] directly compared 20 mg∙kg^−1^∙d^−1^ (~4.2 mg∙d^−1^ total) and 300 mg∙kg^−1^∙d^−1^ (~63.6 mg∙d^−1^ total) doses of HT within a ten-week exercise training program (daily ~65 min run at 75% maximal velocity) and demonstrated that despite a 15% reduction in lipid peroxidation (measured via plasma hyperoxide levels) and 20 mg∙kg∙d^−1^ reduced exercise capacity (measured via the performed physical workload during each training session), there was no impact on the maximal velocity after 10 weeks of supplementation. Conversely, Feng et al. (2011) demonstrated a positive anti-oxidative effect of HT at a similar supplementation level, outlining that 25 mg∙kg^−1^∙d^−1^ of HT in exercised rats (treadmill run 20 m∙min^−1^ and a grade of 5° for 60 min daily, six days a week at randomized times of day) blunts the ROS-dependent autophagic response to exercise, therefore enhancing exercise capacity (defined in this setting as the inability to continue running, and, consequently, the failure to avoid sound and light irritation [40]) and, importantly, antioxidant defenses under exercise conditions.

Further to this, Al Fazazi et al. (2018) [39] provides evidence of the positive impact that 300 mg∙kg^−1^∙d^−1^ can have on exercise performance and exercise capacity, although attributes this to a pro-oxidative effect of HT. Exercise capacity was significantly improved compared to the baseline (week 0) and mid-trial (week 5) within the group (*p* < 0.05), as well as post intervention (week 10) between groups (non-exercise group supplementing 20 mg∙kg^−1^∙d^−1^) (*p* < 0.05); however, this has been attributed to a pro-oxidative effect of HT. Although not confirmed, a possible explanation for this could be due to the training undertaken by the rats in this study (treadmill run at ~75% $\overset{\cdot }{V}$O_2max_ for ~25–65 min∙d^−1^). Interestingly, Al Fazazi et al. (2018) [39] also reported a significant group × time interaction (*p* = 0.038, eta^2^ = 0.155, 1 − β = 0.622) when analyzing the physical work ((J) = force × vertical distance, where force = body weight (kg) × 9.8 m/s^2^, and vertical distance = speed (m/min) × time (min)) performed in each training session. Exercised rats consuming 300 mg∙kg^−1^∙d^−1^ increased their physical work from the first half of the intervention (baseline to week 5) to the second half (week 6 to week 10) (*p* < 0.05), whereas the lower supplementation dose (20 mg∙kg^−1^∙d^−1^) significantly decreased the workload (*p* < 0.05). It is understood that a greater oxidative, and therefore inflammatory response to exercise training, provides a superior adaptation, in comparison to those who experience a lower level of oxidative stress [45]. Despite the relative context of this animal model, due to the mechanistic action of exogenous polyphenol intake, a similar outcome may also be observed in a human model when exercise training is consistent (~65 min daily at ~75% $\overset{\cdot }{V}$O_2max_), although this hypothesis has not yet been explored. The potential for polyphenols to become a pro-oxidant has been previously described with quercetin and outlines the important role of glutathione concentration on the oxidative reaction of polyphenols [46]. Should the glutathione concentration be too low (relative values on an individual basis), the oxidative reactions of polyphenols, in this case HT, may concentrate on protein thiols and in turn, render the production of various HT metabolites pro-oxidant [46]. This too is plausible in the presence of a high polyphenol dose (such as 300 mg∙kg^−1^∙d^−1^ HT) in conjunction with exercise or intense physical activity, as any adaptations to exercise are a result of relative increases in oxidative stress and, in most cases, an adjunct inflammatory response [39].

The broad ranges in the exhibited dose response mirror the wider field of antioxidants and polyphenols in exercise-induced inflammation. However, the disturbed redox balance present with exercise-induced oxidative stress is a direct result of reactive oxygen species’ generation exceeding the endogenous antioxidant capacity. Mikami et al. (2021) [41] detail the impact of oleuropein-rich olive leaf extract, on a battery of cognitive and behavioral tests (sucrose preference test, forced swimming test, and contextual fear-conditioning test) in addition to physiological parameters (time to exhaustion run at 20 m∙min^−1^ and 10° incline at atmospheric (21% O_2_) hypoxic conditions (16% O_2_)) in C57BL/6J mice. With evidence of potent anti-inflammatory actions [47], the inclusion of oleuropein elicited a superior exercise capacity at atmospheric (*p* < 0.05) and hypoxic conditions (*p* < 0.01), equating to a greater percentage of improvements in time to exhaustion run times (~10% compared to ~40%). With the understanding that hypoxic conditions interrupt oxidative phosphorylation [48], adequate ATP production will therefore rely upon an increase in mitochondrial mass, facilitated by the activation of PGC-1α, following an increased consumption of ODP. Although to date there has not been an investigation using a human model, this evidence, for the first time, highlights the possible impact that ODP could have in hypoxic conditions, and furthermore, warrants an investigation into the possible vasodilatory capacity of ODP in the same manner that grape polyphenols (some of which, such as HT, also occur in olives) potentiate vasodilation [49]. Additional investigation into this area may have pertinent application to athletes and exercising individuals, as well as the health of the broader population due to the decreased resistance in vascular structure and subsequent increase in blood flow [50].

### 4.2. Human Studies

Chronic supplementation was the most common strategy utilized within human studies [Roberts et al. (2022), Shirai et al. (2023)] [34,42], while Esquius et al. (2021) [37] employed an acute dose of 25 mL extra virgin olive oil both pre and during exercise (graded maximum exercise test and 45 min maximum distance run). Despite using an acute dose of relatively low ODP (~31.5 mg [51]), supplementation resulted in a 46% reduction in myeloid dendritic cells (MDCs) 24 h following exercise. It is understood that dendritic cells (DCs) are susceptible to cytokine-mediated activation in addition to producing inflammatory cytokines such as IL-10 and IL-12 in both circulation and tissue [52]. This, coupled with the knowledge that DCs are a bridge between innate and adaptive immunity and that MDCs can differentiate into monocytes and macrophages [53], provides evidence of the association with inflammation that perhaps warrants further investigation despite the clear challenges within accurately identifying the derivation of certain cytokines. The increase in MDCs following intense exercise has been reported elsewhere [54,55]; however, during moderate (4-week running training block) and strenuous exercise (marathon run), a polyphenol-rich diet did not significantly mobilize DCs. Therefore, with the understanding that ODPs possess protective and antioxidative effects [24,56], it could be inferred that olive-specific polyphenols, in particular HT, are responsible for the exhibited reduction in immunosuppression following maximal running bouts. However, further research identifying the role of the NF-κB- and mitogen-activated protein kinase (MAPK) signaling pathways in DCs and MDCs’ modulation is necessary to ascertain the impact at a muscular level during various exercise intensities and further understand how MDCs may impact exercise-induced inflammation.

In both studies that implemented chronic supplementation, there is evidence of an association between ODP and the physiological responses to exercise-induced inflammation. Roberts et al. (2022) [34] demonstrated a modest antioxidant effect of 57.6 mg∙d^−1^ of HT in commercially available polyphenol-rich olive fruit water (OliPhenolia^®^). Sixteen days of supplementation increased glutathione (GSH) post exercise and reduced superoxide dismutase (SOD) activity immediately and 24 h following a fatiguing running protocol [34]. The suppression of SOD via the conversion of superoxide anion (O_2_^•−^) to hydrogen peroxide (H_2_O_2_) and oxygen (O_2_) may be indicative of a lower oxidative stress response, which despite the debate in the literature [57], aligns with the suggested scavenging and anti-inflammatory effects of HT (the most potent polyphenol in the supplemented phytocomplex) on O_2_^•−^ [58]. Further investigation into whether HT (in isolated form or from wastewater sources) can impact the mitochondria in muscle cells is required to accurately ascertain where the impact from HT emerges, i.e., muscle tissue or in fact further upstream at the gut level. In addition, the potent scavenging potential of HT may extend to H_2_O_2_ [6,10], and therefore, through reduced hydroxyl (^•^OH) production, support acute recovery from aerobic exercise via a reduction in mitochondrial or DNA damage [34]. Interestingly, Feng et al. (2011) [40] reported that exercise-induced increases in manganese super oxide dismutase (MnSOD) were significantly inhibited following 25 mg∙kg^−1^∙d^−1^ HT supplementation for eight weeks (see Table 2). As a mitochondrial antioxidant enzyme, MnSOD protects cells from oxidative damage by converting superoxide into O_2_ and H_2_O_2_ [59,60]. This mechanism, imposed by HT supplementation, could prove pertinent to both single and repeat exercise bouts, as lower levels of circulating MnSOD following exercise can be considered as an indicator of a reduction in oxidative stress and may therefore support recovery from exercise at a muscular level. It could be proposed that for people who are undertaking unaccustomed exercise, or are new to exercise training, the possible improvements in acute recovery will support a level of consistency that may facilitate longer term adaptations to exercise. In order to empirically corroborate these hypotheses, further research within the specific area of ODPs and exercise-induced inflammation is required.

Shirai et al. (2023) [42] was the only included literature that employed team-based athletes, allowing for sport-specific exercise protocols to be employed and therefore provide insights into possible real-world exercise-based applications of olive-derived nutrients. It was reported that seven days of supplementation of 60 mg maslinic acid (a bioactive pentacyclic triterpenoid found in olives (chemical formula: C_30_H_48_O_4,_ molar mass: 472.70 g·mol^−1^)) reduced perceptual whole body muscle fatigue (*p* < 0.05) and muscle soreness (whole body and individual body parts (shoulder, chest, lower back, and femur)) (*p* < 0.05) in trained water polo athletes. Although no marked improvement in sport-specific performance was reported, it is noteworthy that this study involved trained athletes, completing ~12 sessions of high intensity activity a week, and thus, the athletes may be expected to present a relatively low inflammatory response to the sport-specific exercise protocols [42]. Perhaps, more pertinently, however, was the rate of change in inflammation (TNF-α (*p* < 0.05), high sensitivity C-reactive protein (*p* < 0.05), and oxidative stress-related protein expression, such as cyclooxygenase-2 (COX-2) and NF-κB (*p* < 0.05), in the maslinic acid condition when compared to a placebo. Because of the reduction in both TNF-α and thiobarbituric acid reactive substances (TBARSs) in serum, the authors suggested that maslinic acid may combat the accumulation of inflammatory factors and therefore oxidative damage through suppressing their expression. Further to the reported impact of maslinic acid, other ODPs, such as HT from olive vegetation water, have provided in vitro evidence of a similar ability to diminish the secretion and production of inflammatory cytokines and chemokines in macrophages via mediating the NF-κB pathway [61]. In the context of this review, these are two important findings that corroborate wider research in the area of polyphenols and exercise-induced inflammation, and therefore warrant further exploration into ODP. Further investigation into alternate exercise modalities (e.g., cycling or running to account for increased metabolic variability) and specific sports (combat or contact, whereby a greater inflammatory response would be expected) is needed, in addition to alternate duration and dosing strategies to allow for the consideration of an untrained population, and if maslinic acid (or alternate ODP) could support performance measures in conjunction with the reported reduction in inflammation.

Nutrients, such maslinic acid, HT, and other ODPs that inhibit the production of TNF-α, or suppress COX-2, can be considered beneficial in the management of inflammation from a recovery and return to exercise perspective [62,63]. It is, however, understood that the mechanisms underpinning adaptation to exercise require a stressor (volume specific to each individual) to elicit improvements in performance. This notion presents a fine balance between inflammation, that may inhibit subsequent exercise bouts, and not enough of a stressor to support muscular adaptation and therefore improvements in performance. This topic is often overlooked within the literature; however, in line with the reports included in this review, polyphenols, and in this case, ODP, could be considered to facilitate an improved recovery through a reduced time taken to return to baseline inflammatory levels, or decrease the impact of oxidative damage following exercise-induced inflammation.

### 4.3. Observed Limitations of Current Research

A clear limitation when considering the impact of ODP on exercise-induced inflammation is the limited quantity of pertinent or scientifically robust research available. Despite many studies outlining the possible implications of HT and other ODPs on disease states [24], the differences in cell type, immune response, and the presentation and duration of symptoms, between chronic inflammation and acute exercise-induced inflammation, vary significantly [64]. The limited research within this area is reflected in the observed discrepancy between the results in animal and human models following chronic HT supplementation. Although this may be attributed to the differences in doses provided within human and animal models, if deductions are to be made within the context of exercise-induced inflammation, relative and consistent doses within the human no-observed-adverse-effect level (NOAEL) of 50 mg∙kg ^−1^ [65] should be implemented. Further to this, when analyzing the research accepted for this review, various markers of inflammation and oxidative stress are observed. It is therefore challenging to accurately compare findings that may inform important research questions surrounding dose or time course, particularly when proxy measures of key variables are often utilized in human studies, e.g., protein expression (PGC-1α) as an indicator of mitochondrial biogenesis. It is understood that this is also a limitation of human research in the wider field; however, these challenges are elevated in this setting due to the low number of relevant research.

### 4.4. Recommendations for Future Research

This scoping review highlighted that in the context of ODP, pertinent to exercise-induced inflammation, research is in its infancy and therefore the primary recommendation of this review is to develop the current knowledge surrounding the functional effects of ODP on inflammation. Initially, an investigation into the bioavailability of a standardized dose of ODP is warranted. Despite the complex metabolism of polyphenols, additional specific assessments of the absorption and metabolism of standardized ODP doses (and their secondary metabolites) are required to direct dose and time course interactions, and therefore understand how effective ODP supplementation (or specific commercial ODP supplements) may be. With this knowledge, a more comprehensive evaluation of ODPs and exercise-induced inflammation can be undertaken. Building upon the foundation of animal models within the current topic, additional human research that targets HT either in isolation or as a phytocomplex would significantly enhance the current field of research, providing a consistent and pragmatic evaluation of inflammatory and oxidative stress markers. Furthermore, the reviewed research regularly investigates oxidative stress in conjunction with inflammation; thus, future exercise-based studies should also consider integrating both inflammatory and oxidative stress markers in order to confirm the mechanisms of exercise-induced inflammation and better understand the consequential interactions of ODPs.

Finally, it may be pertinent for prospective research to consider analyzing beyond the well-reported single measures of inflammation (e.g., IL-6, TNF-α), and in the future look to analyze inflammatory arrays to assist in expanding the knowledge of where ODPs interact with an inflammatory-inducing stimulus, such as exercise. It is important to understand the topic of exercise-induced inflammation, and as a more robust collection of empirical evidence is required to substantiate a claim, it would therefore be irresponsible to recommend a suggested supplementation strategy (dose or duration) for ODP based off the current literature.

## 5. Conclusions

The findings from this review identify the limited pertinent research within the area of exercise-induced inflammation and ODPs. Animal models provide evidence that consistent HT supplementation (≥2 weeks) in conjunction with chronic exercise may induce an anti-inflammatory response conducive of tissue adaptation. Furthermore, there is a limited consensus on the direction of isolated HT in human models; however, the supplementation of ODPs for ≤2 weeks whilst undertaking acute single exercise sessions was shown to be effective. It is recommended that prospective research investigates the dose and time course interactions of ODPs to accurately inform future supplementation strategies to support acute exercise recovery and adaptations.

## Figures and Tables

**Figure 1 nutrients-17-00223-f001:**
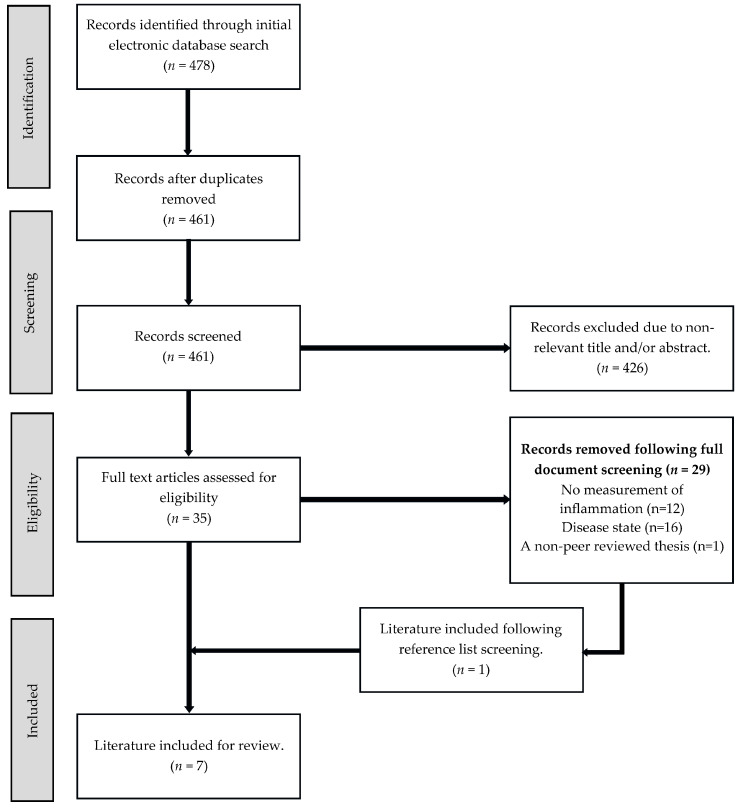
Preferred Reporting Items for Systematic Reviews and Meta-Analyses extension for Scoping Reviews (PRISMA-ScR) flow chart describing the literature screening process.

**Table 1 nutrients-17-00223-t001:** Search terms and MeSH terms utilized for electronic database searches.

Electronic Search Engine	Search Strings
PubMed	(Olive*[ALL] OR Olive* Polyphenol* [ALL] OR hydroxytyrosol [ALL]) AND (exercise AND inflammat*) Olive [ALL]: “olea”[MeSH Terms] OR “olea”[All Fields] OR “olives”[All Fields] OR “olivary nucleus”[MeSH Terms] OR (“olivary”[All Fields] AND “nucleus”[All Fields]) OR “olivary nucleus”[All Fields] OR “olive”[All Fields] Polyphenols [ALL]: “polyphenol’s”[All Fields] OR “polyphenoles”[All Fields] OR “polyphenolic”[All Fields] OR “polyphenolics”[All Fields] OR “polyphenols”[MeSH Terms] OR “polyphenols”[All Fields] OR “polyphenol”[All Fields] hydroxytyrosol [ALL]: “3,4-dihydroxyphenylethanol”[Supplementary Concept] OR “3,4-dihydroxyphenylethanol”[All Fields] OR “hydroxytyrosol”[All Fields] exercise: “exercise”[MeSH Terms] OR “exercise”[All Fields] OR “exercises”[All Fields] OR “exercise therapy”[MeSH Terms] OR (“exercise”[All Fields] AND “therapy”[All Fields]) OR “exercise therapy”[All Fields] OR “exercising”[All Fields] OR “exercise’s”[All Fields] OR “exercised”[All Fields] OR “exerciser”[All Fields] OR “exercisers”[All Fields] inflammation: “inflammation”[MeSH Terms] OR “inflammation”[All Fields] OR “inflammations”[All Fields] OR “inflammation’s”[All Fields]
EBSCOhost	(Olives OR Olive Polyphenols OR hydroxytyrosol) AND (Exercise AND Inflammation)

* A truncation within a PubMed search.

**Table 2 nutrients-17-00223-t002:** Summary of results.

Study	Supplementation	Subjects	Exercise Protocol and Primary Performance Measure	Impact of Supplementation
Animal studies				
Xiong et al., 2022. [38]	Hydroxytyrosol: 25 mg∙kg^−1^∙d^−1^ 45 min before strenuous exercise for 8 weeks. Source: Olive extract powder, DSM Nutritional products, Switzerland.	8-week-old male Sprague Dawley rats weighing 280–300 g.	(a) 60 min fixed speed treadmill run, six days a week for eight weeks. (b) 60 min fixed speed treadmill run.	↓ ROS. ↑ MB.
Al Fazazi et al., 2018. [39]	Hydroxytyrosol 20 mg∙kg^−1^∙d^−1^ (low dose) or 300 mg∙kg^−1^∙d^−1^ (high dose) for ten weeks. Source: Olive extract, Biomaslinic, S.L., Spain.	6-week-old male Wistar rats weighing 212 ± 13.5 g. 10 allocated to exercise with low dose and 10 to exercise with high dose.	(a) Daily treadmill run at a fixed speed. Every other day for five weeks’ duration increased by five minutes (20 min up to 65 min, then was maintained at 65 min), at the end of the five weeks, duration returned to 20 min, and process was repeated for the next five weeks. (b) Graded maximal velocity exercise test.	↑ Lipid peroxidation (high dose). ↑ Maximum running velocity (high dose).
Feng et al., 2011. [40]	Hydroxytyrosol: 25 mg∙kg^−1^∙d^−1^ for eight weeks. Source: Olive extract powder, DSM Nutritional products, Switzerland.	8-week-old male Sprague Dawley rats weighing 180–200 g.	(a) Sixty-minute fixed speed treadmill run, six days a week for eight weeks. (b) Endurance running to exhaustion at a fixed speed.	↑ mitochondrial complex I and II activities. ↑ Endurance capacity. ↓ Exercise-induced renal and immune system damage. ↓ Exercise-induced increased in MnSOD.
Mikami et al., 2021. [41]	‘Oleavita’: Olive leaf extract for five weeks. Source: Ethanol/water olive leaf extract, Phytodia S.A.S., France.	120 10-week-old male C57BL/6J mice, weighing 20–22 g.	(a) Ten minutes of fixed speed treadmill running twice weekly. (b) Endurance running test at fixed treadmill speed in both atmospheric and hypoxic conditions (16% O_2_).	↑ Endurance capacity (atmospheric and hypoxic conditions). ↑ PGC-1α mRNA. ↑ Sirt1 mRNA. ↑ Antioxidant capacity. ↑Anti-inflammatory activity.
Human studies				
Roberts et al., 2022. [34]	‘OliPhenolia^®^’: ~56 mL a day for sixteen days. Source: Olive fruit water phytocomplex, Fattoria La Vialla, Italy.	15 recreationally active adults (age: 30 ± 2 yrs; body mass: 76.7 ± 3.9 kg; height: 1.77 ± 0.02 m).	(a) Habitual exercise training throughout supplementation period (n = 13 ± 1) in addition to laboratory-based 65 min treadmill run at a constant speed (~75% $\overset{\cdot }{V}$O_2max_). (b) A submaximal incremental exercise protocol, ten-minute recovery, followed by maximal test to volitional exhaustion.	↓ SOD activity immediately post exercise and 24 h post exercise. ↑ in GSH immediately post exercise.
Shirai et al., 2023. [42]	Maslinic acid: 60 mg daily for seven days (gel form). Source: Olive fruit extract, NIPPIN CORPORATION, Japan.	12 male national level water polo athletes.	(a) Habitual training eleven times a week. (b) Interval exercise (twenty-second eggbeater holding a 10 kg weight kick + twenty-second rest) was performed as one set until exhaustion.	↓ Perceptual fatigue and muscle soreness during habitual training. ↓ TNF-α. ↓ Creatinine.
Esquius et al., 2021. [37]	Extra virgin olive oil: 25 mL contained within 100 mL commercial orange juice and 8 g modified starch (gel form). Source: Not disclosed.	3 physically active men.	(a) Habitual exercise training ~3–5 times weekly. (b) Graded maximal exercise test to volitional exhaustion, followed by a 5 min rest and 45 min maximum distance run.	↓ mDC 24 after exercise.

SOD = superoxide dismutase, GSH = glutathione, TNF-α = tumor necrosis factor alpha, mDCs = myeloid dendritic cells, ROS = reactive oxygen species, MB = mitochondrial biogenesis, O_2_ = oxygen, a = exercise conducted during supplementation period, b = experimental exercise assessment following supplementation period, ↑ = increase, ↓ = decrease.

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
