# Peer review of "The Application of Olive-Derived Polyphenols on Exercise-Induced Inflammation: A Scoping Review"

_nutrients, 2025, doi:10.3390/nu17020223_

Round 1
Reviewer 1 Report
Comments and Suggestions for Authors
Thank you for sharing your research.
This scoping review examines the impact of olive-derived polyphenols (ODP), particularly hydroxytyrosol (HT), on exercise-induced inflammation. Seven studies were analyzed, reporting potential reductions in oxidative stress, improvements in inflammatory and immune biomarkers, and modulations in mitochondrial function. The studies used average doses of 100.9 mg∙d⁻¹ of ODP over approximately 40 days, primarily with aerobic exercise. Results for HT in human models are inconsistent, whereas animal models showed reduced inflammation after ≥2 weeks of supplementation.
I found the topic potentially interesting; however, the limitations of your research should be better highlighted, particularly addressing the discrepancy between results in animal and human models.
Regards
Author Response
Reviewer comment: Thank you for sharing your research.
This scoping review examines the impact of olive-derived polyphenols (ODP), particularly hydroxytyrosol (HT), on exercise-induced inflammation. Seven studies were analyzed, reporting potential reductions in oxidative stress, improvements in inflammatory and immune biomarkers, and modulations in mitochondrial function. The studies used average doses of 100.9 mg∙d⁻¹ of ODP over approximately 40 days, primarily with aerobic exercise. Results for HT in human models are inconsistent, whereas animal models showed reduced inflammation after ≥2 weeks of supplementation.
I found the topic potentially interesting; however, the limitations of your research should be better highlighted, particularly addressing the discrepancy between results in animal and human models.
Regards
Authors response: Thank you very much for your review of this manuscript and thank you for describing minor areas for improvement. The authors are in agreement with your propositions and therefore have adjusted the text in order to strengthen the final manuscript. The limitations of the research have been further highlighted in addition to providing a more detailed point that addresses the discrepancy in findings between the animal and human models, discussed (line 411 - 416). “The limited research within this area is reflected in the observed discrepancy between results in animal and human models following chronic HT supplementation. Although this may be attributed to the differences in the average dose provided within human and animal models, if deductions are to be made within the context exercise-induced inflammation, relative and consistent doses within the human no-observed-adverse-effect level (NOAEL) of 50 mg∙kg -1 [66] should be implemented.”
Reviewer 2 Report
Comments and Suggestions for Authors
This paper presents the literature findings of an in vivo study of the effects of olive-derived polyphenols (ODP) on exercise-induced inflammation. The manuscript was well prepared on the basis of data provided in 7 articles selected from more than 400 papers according to PRISMA-ScR rules. A valuable achievement of the authors is also the two subsections included at the end of the paper; 4.3 Observed limitations of current research and 4.4 Recommendations for future research.
However, a weakness of the munuscript is the lack of characterization of the product (olive-derived polyphenols), which was the subject of a scoping review. The authors write about the biological activity of ODP, but devote very little space to characterizing the chemical profile indicating only in lines 47-49 “that ODP including phenolic acids and related derivatives such as hydroxytyrosol (HT) and oleuropein (Ole) [9] found in the olive fruit, leaves, unrefined virgin oil and mill wastewater [10].” It is well known that the biological activity of products of plant origin depends not only on single secondary metabolites, but is also the result of the synergistic action of many components. Therefore, in the Introduction section, the authors should include data compiled from the scientific literature on the chemical composition of the ODP under study. Such data must be found in the literature, since olive oil has been used for centuries.
The primary component of ODP mentioned in the paper is hydroxytyrosol (HT). Also provide a brief description of HT, in particular, the structural formula and basic physicochemical properties, how it was obtained from olive oil or otherwise, was it subjected to a standardization process?
Author Response
Reviewer comment: This paper presents the literature findings of an in vivo study of the effects of olive-derived polyphenols (ODP) on exercise-induced inflammation. The manuscript was well prepared on the basis of data provided in 7 articles selected from more than 400 papers according to PRISMA-ScR rules. A valuable achievement of the authors is also the two subsections included at the end of the paper; 4.3 Observed limitations of current research and 4.4 Recommendations for future research.
However, a weakness of the munuscript is the lack of characterization of the product (olive-derived polyphenols), which was the subject of a scoping review. The authors write about the biological activity of ODP, but devote very little space to characterizing the chemical profile indicating only in lines 47-49 “that ODP including phenolic acids and related derivatives such as hydroxytyrosol (HT) and oleuropein (Ole) [9] found in the olive fruit, leaves, unrefined virgin oil and mill wastewater [10].” It is well known that the biological activity of products of plant origin depends not only on single secondary metabolites, but is also the result of the synergistic action of many components. Therefore, in the Introduction section, the authors should include data compiled from the scientific literature on the chemical composition of the ODP under study. Such data must be found in the literature, since olive oil has been used for centuries.
The primary component of ODP mentioned in the paper is hydroxytyrosol (HT). Also provide a brief description of HT, in particular, the structural formula and basic physicochemical properties, how it was obtained from olive oil or otherwise, was it subjected to a standardization process?
Authors response: Thank you for the detailed review of our manuscript, you raise some excellent points and we, the authors, have adjusted the manuscript to reflect the minor suggestions and therefore strengthen the final manuscript.
Reviewer suggestion 1: ‘Include additional information surrounding the chemical composition of the ODP under study.’ The authors have included the following text (line 46 – 56): “Primarily, olives and olive oil are comprised of saturated (in the form of glycerol esters), poly-unsaturated and mono-unsaturated fatty acids in addition to a variety of vitamins and minerals (Vitamin K, tocopherols, beta-carotene, calcium, iron and potassium) in different quantities and ratios depending upon the strain and harvest of olive fruit [9]. In addition to tocopherols (Vitamin E), various other antioxidants and polyphenols are present within olives, olive leaves and olive oil [10]. The most potent concentration is provided by secoiridoids such as oleacein, oleocanthal and oleuropein aglycone, however lignans, flavones and phenolic acids and alcohols are also present and may interact to elicit physiological effects, although concentrations vary greatly between crop, strain and part of the fruit. [10,11]”. In addition to including the chemical formula and molar mass of the primary ODPs under study within the records included for this review (Lines 57, 58 and 368): “HT: C8H10O3, molar mass: 154.17 g·mol−1, Ole: C25H32O13 molar mass: 540.52 g·mol−1, MA: C30H48O4 molar mass: 472.70 g·mol−1”.
Reviewer suggestion 2: ‘Provide a brief description of HT, in particular, the structural formula and basic physicochemical properties, how it was obtained from olive oil or otherwise, was it subjected to a standardization process’. In response to this excellent point, the authors have included commentary of the physiochemical and biological properties of HT throughout the text (including the chemical formula and molar mass form the above suggestion (Lines 51, 52 and 370)). Additionally the authors have included a brief explanation surrounding the extraction of hydroxytyrosol (HT), which we agree, will provide further clarity and therefore strengthen the current manuscript (Lines 62 - 65): “In most cases, HT is extracted from olive leaves via a four-step process; increasing the area of the olive leaf (pre-treatment), solid-liquid extraction, hydrolysis reaction and pH adjustment [18] however, other methods have been described elsewhere [19]”.
Reviewer 3 Report
Comments and Suggestions for Authors
The aim of the current scoping review is to understand the impact of olive derived polyphenols may have on exercise-induced inflammation. The literature searches were conducted in PubMed and EBSCO host and considered for review if records reported original data, examined olives, olive derived nutrients, food sources or ODP in conjunction with exercise-induced inflammation.
Some comments/suggestions:
1. You wrote about hydroxytyrosol (HT) and oleuropein (Ole) as derived polyphenols in olive. Please specify if there are other derived polyphenols in olives and add the chemical structures for the compounds.
2. Page 2, lines 58-63, you described the metabolism of HT. Please add a schema with the chemical reactions. For Ole are known more details regarding metabolism?
3. What other active principles are found in olives? Can't they have the same action with ODP? Please clarify.
4. Page 4, lines 159-161, you wrote: “all literature that included a clear disease state (e.g., cardiovascular disease, diabetes mellitus, Alzheimer’s disease) were discarded regardless of links to inflammation”. Please specify exactly all the diseases that represented exclusion criteria. Other diseases are responsible also for inflammation.
5. Table 2 – You must add the origin of HT, Oleavita, OliPhenolia, Maslinic acid, Extravirgin olive oil. All of them where administrated orally? 6. Were positive controls used for the in vivo studies? Please add/clarify.
Author Response
Reviewer comment: The aim of the current scoping review is to understand the impact of olive derived polyphenols may have on exercise-induced inflammation. The literature searches were conducted in PubMed and EBSCO host and considered for review if records reported original data, examined olives, olive derived nutrients, food sources or ODP in conjunction with exercise-induced inflammation.
Some comments/suggestions:
- You wrote about hydroxytyrosol (HT) and oleuropein (Ole) as derived polyphenols in olive. Please specify if there are other derived polyphenols in olives and add the chemical structures for the compounds.
- Page 2, lines 58-63, you described the metabolism of HT. Please add a schema with the chemical reactions. For Ole are known more details regarding metabolism?
- What other active principles are found in olives? Can't they have the same action with ODP? Please clarify.
- Page 4, lines 159-161, you wrote: “all literature that included a clear disease state (e.g., cardiovascular disease, diabetes mellitus, Alzheimer’s disease) were discarded regardless of links to inflammation”. Please specify exactly all the diseases that represented exclusion criteria. Other diseases are responsible also for inflammation.
- Table 2 – You must add the origin of HT, Oleavita, OliPhenolia, Maslinic acid, Extravirgin olive oil. All of them where administrated orally?
- Were positive controls used for the in vivo studies? Please add/clarify.
Authors response: Thank you for your detailed review of our manuscript. The authors have taken onboard the minor suggestions and comments you have made and agree that the inclusion of the below edits will greatly strengthen this manuscript. The authors have responded point by point as the comments were originally made.
- Thank you for highlighting this, the authors have now added further information to outline other polyphenols present within olives as well as their potency (lines 50 – 56) and included the chemical structure of the assessed ODP, on lines 57, 58 and 368.
- Thank you for your comment, upon discussion with the research team propose that this is more appropriate and can be achieved in text and have therefore included lines 71 - 77 with supporting references [21 - 23] to concisely reflect this.
- Thank you for highlighting this point, in response to this, the authors have included the following text (line 46 – 56): to provide additional insights into the composition of the ODP under investigation. “Primarily, olives and olive oil are comprised of saturated (in the form of glycerol esters), poly-unsaturated and mono-unsaturated fatty acids in addition to a variety of vitamins and minerals (Vitamin K, tocopherols, beta-carotene, calcium, iron and potassium) in different quantities and ratios depending upon the strain and harvest of olive fruit [9]. In addition to tocopherols (Vitamin E), various other antioxidants and polyphenols are present within olives, olive leaves and olive oil [10]. The most potent concentration is provided by secoiridoids such as oleacein, oleocanthal and oleuropein aglycone, however lignans, flavones and phenolic acids and alcohols are also present and may interact to elicit physiological effects, although concentrations vary greatly between crop, strain and part of the fruit. [10,11]”.
- Thank you for commenting on this, upon reflection the authors agree this can be made clearer to more accurately reflect the exclusion process (i.e., all records that included investigation into disease states were excluded). Lines 156 - 158 have now been adapted and read “Importantly at this stage, all records that included investigation into a disease state were discarded”. The authors believe this now clearly explains that all records that examine a disease state were removed and therefore listing all possible diseases that may be linked to inflammation is not warranted.
- Thank you for raising this point, in response to this, the authors have now adjusted Table 2 and added the origin and source of all supplemented products investigated within the records included in this review. The authors can confirm that all supplementation was administered orally.
- Thank you for raising this important question. The authors can confirm that all records included in this review included placebo controls. This is now covered in line 189 – 190 “and importantly all records included within this review employed an experimental control” to further emphasise this point.